# A Method for Improving Controlling Factors Based on Information Fusion for Debris Flow Susceptibility Mapping: A Case Study in Jilin Province, China

**DOI:** 10.3390/e21070695

**Published:** 2019-07-15

**Authors:** Qiang Dou, Shengwu Qin, Yichen Zhang, Zhongjun Ma, Junjun Chen, Shuangshuang Qiao, Xiuyu Hu, Fei Liu

**Affiliations:** 1College of Construction Engineering, Jilin University, Changchun 130026, China; 2Jilin Institute of Geological Environment Monitoring, Changchun 130021, China

**Keywords:** debris flow susceptibility mapping, Jilin province, information fusion, statistical index, analytic hierarchy process, random forest method, Spearman’s rank correlation coefficients

## Abstract

Debris flow is one of the most frequently occurring geological disasters in Jilin province, China, and such disasters often result in the loss of human life and property. The objective of this study is to propose and verify an information fusion (IF) method in order to improve the factors controlling debris flow as well as the accuracy of the debris flow susceptibility map. Nine layers of factors controlling debris flow (i.e., topography, elevation, annual precipitation, distance to water system, slope angle, slope aspect, population density, lithology and vegetation coverage) were taken as the predictors. The controlling factors were improved by using the IF method. Based on the original controlling factors and the improved controlling factors, debris flow susceptibility maps were developed while using the statistical index (SI) model, the analytic hierarchy process (AHP) model, the random forest (RF) model, and their four integrated models. The results were compared using receiver operating characteristic (ROC) curve, and the spatial consistency of the debris flow susceptibility maps was analyzed while using Spearman’s rank correlation coefficients. The results show that the IF method that was used to improve the controlling factors can effectively enhance the performance of the debris flow susceptibility maps, with the IF-SI-RF model exhibiting the best performance in terms of debris flow susceptibility mapping.

## 1. Introduction

A fast-moving debris flow that has a wide influence range can be defined as a transient mass motion within the loose steep slope channel due to rainfall. Debris flow, which causes a substantial loss of lives and property [1,2], has become one of the most dangerous geological disasters in the world and it poses a serious threat to the living environment of humans [3].

Debris flow susceptibility mapping (DFSM) can predict the location of debris flow that is based on terrain, as well as geological and hydrological characteristics, to prevent and reduce the impact of debris flow disasters [4,5,6]. With the development of GIS and remote sensing technology an increasing number of methods are used for DFSM.

The models for DFSM in previous studies are mainly divided into three categories: statistical models, heuristic models, and soft computing models [7]. For statistical models, Xu et al. [8] used the information value model to analyze the debris flow susceptibility in Sichuan province, China. In addition, the logistic regression model [9], evidence belief function [10], weight of evidence [11], frequency ratio [10,12], and statistical index (SI) [13] have been used extensively. Regarding heuristic models, Addison et al. [14] used the classifier tree to analyze debris flow. Other models, such as multistandard analysis [15] and the heuristic fuzzy model [16], have also been used extensively. For soft computing models, artificial neural networks [17,18,19], maximum entropy [20], decision tree [21], and classification and regression trees [22] are equally common. 

However, research on the selection of controlling factors, which are the foundation of DFSM and affect the accuracy of debris flow susceptibility maps, is still rare. Previous studies have focused on converting a group of controlling factors that may be relevant into a set of linear unrelated controlling factors while using principal component analysis [23,24,25,26,27]. However, this method will reduce the number of controlling factors that are used in the study, resulting in a reduction in the utilization of the underlying data, which will eventually have a negative impact on the accuracy of the final debris flow susceptibility map. 

Therefore, an information fusion (IF) method that is based on the Minkovsky distance [28,29] and the Dempster-Shafer theory [30,31,32] was proposed in this study to improve the rationality of the controlling factors, the utilization of the underlying data, and the accuracy of the final debris flow susceptibility map. Based on the original controlling factors and the improved controlling factors, debris flow susceptibility maps were developed while using the selected models and compared using the receiver operating characteristic (ROC) curve. To comprehensively verify the applicability of the IF method in different types of models, the SI model was selected from the statistical models, the analytic hierarchy process (AHP) model was chosen from the heuristic models, and the random forest (RF) method was selected from the soft calculation models. Then, these three models were integrated to obtain four integrated models, that is, the combination of SI and AHP (SI-AHP), the combination of SI and RF (SI-RF), the combination of AHP and RF (AHP-RF), and the combination of SI, AHP, and RF (SI-AHP-RF). Finally, to prove the superiority of the IF method, the spatial consistency of the debris flow susceptibility maps was analyzed while using Spearman’s rank correlation coefficients.

## 2. Study Area

The research area (Jilin province) is located in the northeastern part of China (Figure 1), from approximately 40°52′N to 46°18′N and from 121°38′E to 131°19′E, extending along the northeast-southwest direction, and with a total area of 18.74 km^2^. The area is in the eastern monsoon climate zone of China, and it has an annual average temperature between 2 and 6 °C and annual precipitation in the range from 400–1000 mm, both of which gradually decrease from southeast to northwest. The precipitation in the summer accounts for 70%–80% of the annual precipitation. 

The study area is 5.0 m to 2691 m above sea level, and it is located in two geomorphological units: the eastern Changbai Mountain and the western Songliao Plain. The terrain is high in the southeast and low in the northwest. Bounded by the latitude 42°40′–43°, the study area spans two major structural units: the Tarimand-China-north Korea para platform area and the Tianshan-Xingan trough fold area.

The exposed strata are from the Archean Eon to the Cenozoic Era. The rock mass can be divided into 13 rock groups, according to lithological characteristics. The lithology consists mainly of granite, basalt, glutenite, clay rock, pyroclastic rock, carbonate rock, and gneiss.

The geological environmental conditions in the study area are complex, and the occurrence of geological hazards is the result of multiple factors. After the study area enters the rainy season, debris flow occurs frequently, causing huge economic losses every year. We selected Jilin province as the study area, because such disasters occur frequently in this region; thus, sufficient data can be collected to verify the IF method, which has practical value for our research.

## 3. Data Preparation

Data collection is the basis for subsequent analysis [33,34,35]. In this study, a debris flow inventory map, including 868 debris flow events, which was compiled based on the debris flow data before 2012, as provided by the Jilin Provincial Department of Land and Resources, was combined with field investigation data (Figure 2). Then, the debris flow events were randomly divided into training and validation datasets: 70% (608 events) were used for training the models, and 30% (260 events) were used for validation.

According to the characteristics of debris flow and the results of a 1:100,000 geological disaster investigation in the study area, the relationship between debris flow and the geological environment was analyzed. Then, based on experience [36,37,38,39,40,41], nine layers of debris flow controlling factors (i.e., topography, elevation, annual precipitation, distance to water system, slope angle, slope aspect, population density, lithology, and vegetation coverage) were taken as predictors. The spatial database for the study area is shown in Table 1. 

The details of the grading standard of each controlling factor are as follows. The hierarchical diagram of the controlling factors is shown in Figure 3.

Topography affects the formation, movement, and scale of debris flow [42]. In this study, according to the order of the density of debris flow point in each geomorphic unit from small to large, the topography was divided into group 1 (Songliao low plain, piedmont slope plain, high plain), group 2 (mountain basin, valley), group 3 (hills, low hills), and group 4 (medium and low mountains, terraces).

The relative height difference determines the gravitational potential energy inside the slope [3]. According to survey statistics, the Changbai Mountain area in the eastern part of the study area is 800–1500 m above sea level: the main peak of Changbai Mountain and its surrounding peaks are above 2000 m, and the central plateau plain is 600–800 m above sea level. According to these critical values, the elevation was divided into four classes: 0–600 m, 600–800 m, 800–1500 m, and >1500 m.

Rainfall plays an important role in slope instability [4]. Debris flow disasters are mostly caused by continuous rainfall. Therefore, the annual precipitation, which is mainly distributed between 600–1000 mm in the study area, was selected as the controlling factor [43,44]. Then, the interval from 600–1000 mm was divided into two parts on average. Thus, annual precipitation was divided into four categories: 0–600 mm, 600–800 mm, 800–1000 mm, and >1000 mm.

The steep slopes provide loose material for debris flow [45]. The slope angle in the northwest of the study area is mainly 0–5°, while the slope in the southeast is generally near 10°, and, in a few areas, it is more than 20°. According to these critical values, the slope angle was divided, as follows: 0–5°, 5–10°, 10–20°, and >20°.

The slope aspect is related to precipitation and topographical trends [1]. According to the influence of light, the slope aspect was divided into shady slope (135°–180°, 180°–225°), semi-shady slope (45°–90°, 270°–315°), semi-sunny slope (90°–135°, 225°–270°), and sunny slope (135°–225°).

The population density indirectly reflects the influence of human activities on the geological environment. Human activities can cause vegetation degradation and changes in topography and geomorphology, indirectly increasing the possibility of debris flow. According to the number of people per square kilometers, the population density was divided into four classes: very low (0–10), low (10–100), moderate (100–500), and high (>500).

The lithology controls the stability of the slope and determines the amount of material that is available for debris flow [46,47]. According to the anti-weathering ability of rock, the lithology was divided into four types: soil, soft rock, hard rock and extremely hard rock.

The lower the vegetation coverage is, the more easily the rock mass becomes weathered, and the more likely it is that debris flow will occur. In this study, the vegetation coverage of the eastern Changbai Mountains is more than 80%, while that of the western plain is less than 20%. The vegetation coverage in the central region is mainly between 20% and 50%. According to these critical values, vegetation coverage was divided into four classes: low (<20%), moderate (20%–50%), high (50%–80%), and very high (>80%).

Rivers will erode the rock mass at the bottom of a slope, affecting the stability of the slope. In general, the likelihood of debris flow decreases as the distance to the water system increases [13]. In this study, the distance to the water system was divided into six classes:0–500 m, 500–1000 m, 1000–1500 m, 1500–2000 m, 2000–2500 m, and >2500 m.

## 4. Methodology

The methods that were used in this study can be summarized in three parts. The first part is the IF method for improving the controlling factors. The second part is the models for DFSM. Six models, i.e., SI, AHP, SI-AHP, SI-RF, AHP-RF, and SI-AHP-RF, were used to develop the debris flow susceptibility maps. The third part is the method that is used to verify the results. ROC curve analysis is used to verify the success rate and prediction rate of the debris flow susceptibility maps, and the Spearman’s rank correlation coefficients are used to verify the spatial consistency of the debris flow susceptibility maps. Figure 4 shows the flow of the research methods.

### 4.1. The Information Fusion Method

The IF method is based on the Minkowski distance and Dempster-Shafer theory. The Minkowski distance, which is a distance function that is defined on eigenvector space [48,49] is used to measure the similarity between the controlling factors. The Dempster-Shafer theory is used to calculate the credibility degree of each controlling factor. The credibility degree is used as a weight to improve the layer of each controlling factor, and it is the result of the IF method. The calculation process of the credibility degree is as follows:

Step 1: Assign the grade of each controlling factor from small to large while using a value from 1 to 4 or a value from 1 to 6. For example, for annual precipitation, the intervals 0–600 mm, 600–800 mm, 800–1000 mm, and >1000 mm was assigned values of 1, 2, 3, and 4, respectively. Finally, the column vector was obtained according to the values of all the disaster points in a controlling factor.

Step 2: Calculate the Minkowski distance according to column vectors of the controlling factors:(1)$$D_{ij}={\left[\overset{n}{\underset{i}{\sum }}{\left|x_{1i}-x_{2i}\right|}^{m}\right]}^{1/m},$$
where $D_{ij}$ is the Minkowski distance between controlling factor *i* and controlling factor *j*;$\;x_{1i}$ and $x_{2i}$ are the values of a disaster point in a column vector; and, *m* is a variable parameter.

Step 3: Obtain the similarity measure Matrix, according to the Minkowski distance.
(2)$$\mathrm{COM}=\left(\begin{array}{ccc} \begin{array}{ccc} 1 & D_{12} & D_{13}\\ D_{21} & 1 & D_{23} \end{array} & \cdots & \begin{array}{c} D_{1n}\\ D_{2n} \end{array}\\ \vdots & \ddots & \vdots \\ \begin{array}{ccc} D_{n1} & D_{n2} & D_{n3} \end{array} & \cdots & 1 \end{array}\right)$$

Step 4: Calculate the support degree of the controlling factors:(3)$$Sup\left(X_{i}\right)=\overset{n}{\underset{j=1,j\neq i}{\sum }}D_{ij}$$
where $Sup\left(X_{i}\right)$ is the support degree of controlling factor *i*; $X_{i}$ is the controlling factor *i*; and *n* is the number of controlling factors.

Step 5: Calculate the credibility degree of controlling factor *i*:(4)$$Cfl\left(X_{i}\right)=\frac{Sup\left(X_{i}\right)}{\sum_{j=1}^{n}Sup\left(X_{j}\right)}$$
where $Cfl\left(X_{i}\right)$ is the credibility degree of the controlling factor *i*; $X_{i}$ is the controlling factor i; $X_{j}$ is the controlling factor *j*;$\;Sup\left(X_{i}\right)$ is the support degree of controlling factor *i*; $Sup\left(X_{j}\right)$ is the support degree of controlling factor *j*; and, *n* is the number of controlling factors.

### 4.2. The Models for DFSM

#### 4.2.1. The Statistical Index Model

The SI model is a binary statistical method, whose result can reflect the weights of the controlling factors. [50,51]. The weights are obtained by the following formula.
(5)$$W_{ij}=\ln\left(\frac{M_{ij}}{M}\right)=\ln\left(\frac{D_{ij}/D_{T}}{P_{ij}/P_{T}}\right),$$
where $W_{ij}$ is the weight of grade *j* in controlling factor *i*; $M_{ij}$ is the debris flow density of grade *j* in controlling factor *i*; M is the total density of debris flow within the map;$\;D_{ij}$ is the number of debris flow events of grade *j* in controlling factor *i*;$\;D_{T}$ is the number of debris flow events in the map; $P_{ij}$ is the number of pixels of grade *j* in controlling factor *i*; and,$\;P_{T}$ is the total number of pixels in the map.

#### 4.2.2. The Analytic Hierarchy Process Model 

The AHP model is a multistandard decision-making process and a common method for determining subjective weight [52,53]. There will be some uncertainty results due to the evaluation of different experts. This model can be described in four steps, as follows:

Step 1: Establish a hierarchical analysis structure model for DFSM.

Step 2: Construct a pairwise comparison matrix:(6)$$A=\left(\begin{array}{ccc} a_{11} & \cdots & a_{1n}\\ \vdots & a_{ij} & \vdots \\ a_{n1} & \cdots & a_{nn} \end{array}\right),$$
where *A* is the pairwise comparison matrix and $a_{ij}$ is the result of comparison between controlling factor *i* and controlling factor *j*.

Step 3: Calculate the weight vector from the pairwise comparison matrix. Determine the weight of each controlling factor.

Step 4: Check the consistency of the weights, and when the consistency ratio (*CR*) is less than or equal to 0.1 the result is considered reasonable.
(7)$$CR=\frac{\left(\lambda_{max}-n\right)/\left(n-1\right)}{RI},$$
where, *CR* is the consistency rate; $\lambda_{max}$ is the weighting and vector mean component; *n* is the number of controlling factors; and, *RI* is the degree of freedom index.

#### 4.2.3. The Random Forest Model 

The RF model, which can analyze the importance of classification features and determine the weight of each controlling factor, is a classification model that is composed of many decision trees [54,55]. The RF model includes two main kinds of algorithms: the GINI index algorithm and the out-of-bag (OOB) error rate replacement algorithm. The GINI index is used to calculate the impurity of nodes to measure the weight. The calculation process is as follows:

Step1: Calculate the GINI index of node *C*:(8)$$GI_{c}=1-\overset{k}{\underset{m=1}{\sum }}P_{ck}{}^{2}$$
where $GI_{c}$ is the GINI index of node *C*; k is the *K* category of node *C*; and,$\;P_{ck}$ is the proportion of category *K* in node *C*.

Step 2: Calculate the importance of factor *j* in node *C*:(9)$$IRF_{ij}{}^{\mathrm{GINI}}=GI_{c}-GI_{l}-GI_{r}$$
where $IRF_{ij}{}^{\mathrm{GINI}}$ is the importance of factor *j* in node *C*; and,$\;GI_{l}$ and $GI_{r}$ represent the GINI values at the two new nodes that branch down.

Step 3: Calculate the weight of controlling factor *j*:(10)$$IRF_{j}=\frac{\sum_{i=1}^{n}IRF_{ij}{}^{\mathrm{GINI}}}{\sum_{s=1}^{m}IRF_{s}}$$
where $IRF_{j}$ is the weight of controlling factor *j*; *n* is the number of decision trees; and, *m* is the number of controlling factors.

#### 4.2.4. The Integrated Model

Integrated models can make up for the shortcomings of individual models because of their ability to solve high-dimensional problems and high identification accuracy [7], which lead to more accurate results. Therefore, in this study, the SI model, the AHP model, and the RF model were integrated to get four integrated models: SI-AHP, SI-RF, AHP-RF, and SI-AHP-RF. The integration of individual models is achieved through the following steps:

Step 1: Obtain the weight of each grade of the controlling factor according to the individual models.

Step 2: Standardize these weights in the data analysis module of Statistical Product and Service Solutions (SPSS) software.

Step 3: Obtain the new weights of each grade of the controlling factors in the integrated model by using the following formula:(11)$$\omega =\omega_{i}+\omega_{j}$$
where ω is the new weights of each grade of the controlling factors in the integrated model and $\omega_{i}\;\mathrm{and}\;\omega_{j}$ are the standardized weights of each grade of the controlling factors in the individual models.

#### 4.2.5. Combination of the IF Method and Six Models

The credibility degree that was obtained by the IF method is regarded as the improved weight of each controlling factor. Then, according to the standardized weight of each grade of the controlling factors obtained by using SI, AHP, SI-AHP, SI-RF, AHP-RF, and SI-AHP-RF models, the improved weights of the controlling factors were obtained by the following formula:(12)$$\omega^{\prime }=Cfl\left(X_{i}\right)\times \omega$$
where $\omega^{\prime }$ is the improved weight of controlling factor *i*, $Cfl\left(X_{i}\right)$ is the credibility degree of controlling factor *i*, and ω is the standardized weight of controlling factor *i* that was obtained by the selected model.

### 4.3. Validation of Debris Flow Susceptibility Maps

To compare the performance of different debris flow susceptibility maps and ensure the reliability of the IF method, the ROC curve and the Spearman’s rank correlation coefficient were selected. 

#### 4.3.1. ROC Curve

The ROC curve is drawn from a series of two-category methods (demarcation values or decision thresholds) and it uses sensitivity as the ordinate and specificity as the abscissa. The area under the curve is between 0.5 and 1. The larger the area is, the better the effect of the model [41].

#### 4.3.2. Spearman’s Rank Correlation Coefficient

Spatial consistency can be interpreted as the similarity of the debris flow susceptibility assessment results in the spatial distribution. The Spearman’s rank correlation coefficient, which is a different nonparametric measure of the correlation of variables, was used to evaluate the spatial consistency between the two different debris flow susceptibility maps. The calculation process is as follows:

Step 1: Obtain the column vectors according to the grade of debris flow susceptibility of all the debris flow points. The grades of debris flow susceptibility were assigned values of 1 to 4 from small to large.

Step 2: Calculate the difference D between two column vectors:(13)$$D=\overset{N}{\underset{i=1}{\sum }}{\left|R\left(X_{i}\right)-R\left(Y_{i}\right)\right|}^{2}$$
where D is the difference between two column vectors; *X* and *Y* are the column vectors that were obtained from different debris flow susceptibility maps; $R\left(X_{i}\right)$ and $R\left(Y_{i}\right)$ are the value of the debris flow susceptibility grade corresponding to a disaster point; and,  N is the number of debris flow points.

Step 3: Calculate the correlation between two debris flow susceptibility maps:(14)$$R_{s}=1-\frac{6\times D}{N\times \left(N^{2}-1\right)}$$
where $R_{s}$ is the Spearman’s rank correlation coefficient; D is the difference between two column vectors; and, N is the number of debris flow points.

## 5. Results

### 5.1. The Results of the Information Fusion Method 

The correlation between the selected controlling factors was expressed in terms of the magnitude of the Minkowski distance value. The smaller the Minkowski distance between the controlling factors, the higher the similarity between them. The support degree and the credibility degree of the selected controlling factors are shown in Table 2. The topography has the highest credibility degree value (0.157), and the lowest credibility degree value is 0.086 for elevation. 

### 5.2. DFSM using Six Models Based on Original Controlling Factors

Based on the original controlling factors, the standardized weights, which are shown in the Table 3, were calculated by the selected six models, i.e., SI, AHP, SI-AHP, SI-RF, AHP-RF, and SI-AHP-RF. The debris flow susceptibility maps (Figure 5) were finally obtained by superimposing the layers of the controlling factors according these weights in the ArcGIS software. The susceptibility of debris flow was divided into four grades—low, moderate, high, and very high—according to the natural fracture method [56]. 

### 5.3. DFSM Using Six Models Based on Improved Controlling Factors

Based on the improved controlling factors, the new standardized weights of controlling factors, which are shown in the Table 4, were calculated by the selected six models i.e., IF-SI, IF-AHP, IF-SI-AHP, IF-SI-RF, IF-AHP-RF, and IF-SI-AHP-RF. The debris flow susceptibility maps (Figure 6) were finally obtained by superimposing the layers of the improved controlling factors, according to these new weights in the ArcGIS software. 

### 5.4. Validation

#### 5.4.1. Results of the ROC Curve

The success rate comes from the training dataset, and the prediction rate comes from the validation dataset. The pairwise comparison results between the models are show in Figure 7; Figure 8 shows, which reveal an improvement in the performance of the models based on the IF method. For the debris flow susceptibility maps based on the original controlling factors, the success rates of SI, AHP, SI-AHP, SI-RF, AHP-RF, and SI-AHP-RF were 0.813, 0.801, 0.843, 0.887, 0.854, and 0.817, and their prediction rates were 0.817, 0.803, 0.853, 0.888, 0.857, and 0.832, respectively. For the debris flow susceptibility maps that were based on the improved controlling factors, the success rates of IF-SI, IF-AHP, IF-SI-AHP, IF-SI-RF, IF-AHP-RF, and IF-SI-AHP-RF were 0.854, 0.856, 0.900, 0.930, 0.905, and 0.870, and their prediction rates were 0.868, 0.866, 0.910, 0.940, 0.922, and 0.903, respectively. These results reveal the better performance of the debris flow susceptibility maps that are based on the improved controlling factors.

#### 5.4.2. Results of Spatial Consistency Analysis

The smaller the Spearman’s rank correlation coefficient is, the greater the spatial consistency between the two debris flow susceptibility maps. As shown in Table 5, the Spearman’s rank correlation coefficients between the debris flow susceptibility maps that were obtained by the same models, such as SI and IF-SI, are obviously smaller than the coefficients between other maps. This phenomenon indicates that there was a high spatial consistency between the debris flow susceptibility maps that were obtained by the same models, which proves that the IF method is indeed effective. In addition, the results show that there is also a high degree of spatial consistency between IF-SI-AHP-RF, IF-SI-RF, and IF-AHP-RF, and low spatial consistency between the remaining maps.

## 6. Discussion 

### 6.1. Comparison of Debris Flow Susceptibility Maps

As shown in Figure 7 and Figure 8, the success rate and the prediction rate of the twelve debris flow susceptibility maps are more than 0.8, which indicates that the debris flow susceptibility maps are credible. The IF-SI-RF model, which has the highest success and prediction rates, can be considered to be the best model. The results of the best IF-SI-RF model show that the area ratios of low, moderate, high, and very high were 37.7%, 21.4%, 25.5%, and 15.4%, respectively, and the areas with high susceptibility are distributed mainly in the middle and low mountain areas in the east of the study area.

The success and prediction rates of debris flow susceptibility maps that are based on the improved controlling factors are significantly better than those that are based on the original controlling factors. As shown in Table 6, the success rates of SI, AHP, SI-AHP, SI-RF, AHP-RF, and SI-AHP-RF increased by 4.1%, 5.5%, 5.7%, 4.3%, 5.1%, and 5.3%, respectively, and the prediction rates increased by 5.1%, 6.3%, 5.7%, 5.2%, 6.5%, and 7.1%, respectively, which proves that the IF method can improve the rationality of the controlling factors. In addition, the results of six different types of models were significantly improved, which shows that the scope of application of the IF method is extensive.

### 6.2. Why the IF Method Can Improve the Controlling Factors

It is also necessary to analyze the reasons why the IF method can improve the controlling factors. First, there is an inevitable correlation between the selected controlling factors, because a controlling factor will have an impact on other controlling factors. For example, in the plain area, the slope angle is relatively small. In addition, when the study area is relatively large, the geological environment conditions are complex and diversified, and the same controlling factors will play different roles in different areas. Thus, there will be conflicts between the controlling factors. The IF method can weaken the correlations and conflicts between the controlling factors. In addition, when the principal component analysis method is used to analyze the controlling factors, the number of controlling factors will be reduced. However, the IF method can make full use of the original data, therefore, it can improve the controlling factors, further improving the performance of the debris flow susceptibility maps.

### 6.3. Spatial Consistency Analysis of Debris Flow Susceptibility Maps

The improvement in the success rate and prediction rate of debris flow susceptibility maps is not enough to show the effectiveness of the IF method. Therefore, the spatial consistency of the debris flow susceptibility maps is further analyzed. When the two debris flow susceptibility maps, which were obtained by the same model that was based on the original controlling factors and the improved controlling factors, show high spatial consistency, the improvement in the controlling factors is persuasive. In contrast, if the spatial consistency varies greatly, the improvement in the controlling factors is incorrect. Therefore, to ensure the reliability of the IF method, it is necessary to test the spatial consistency between the debris flow susceptibility maps. The final analysis results show that there is good spatial consistency between the two debris flow susceptibility maps that were based on the original controlling factors and the improved controlling factors, which further proves that the IF method is effective. 

## 7. Conclusions

The IF method is proposed and verified for DFSM by taking Jilin province, China, as a study area. Based on field investigations and historical data, nine debris flow controlling factors were selected and improved by the IF method. The SI, AHP, SI-AHP, SI-RF, AHP-RF, and SI-AHP-RF models were used to develop debris flow susceptibility maps, and the results were compared and verified. The conclusions are as follows.

The success and prediction rates of debris flow susceptibility maps that are based on improved controlling factors are significantly better than those based on the original controlling factors. In addition, according to the results of spatial consistency analysis, the two debris flow susceptibility maps, which were obtained by the same model based on the original controlling factors and the improved controlling factors, have a high spatial consistency. Therefore, the conclusion that the IF method can improve the controlling factors for DFSM is trustworthy.

Regarding the results of the best IF-SI-RF model, areas with high susceptibility are mainly distributed mainly in middle and low mountain areas in the east of the study area. The results of this study can provide reliable information for the prevention and management of debris flow disasters in the study area and they have significance for reducing and avoiding the losses that are caused by debris flow.

## Figures and Tables

**Figure 1 entropy-21-00695-f001:**
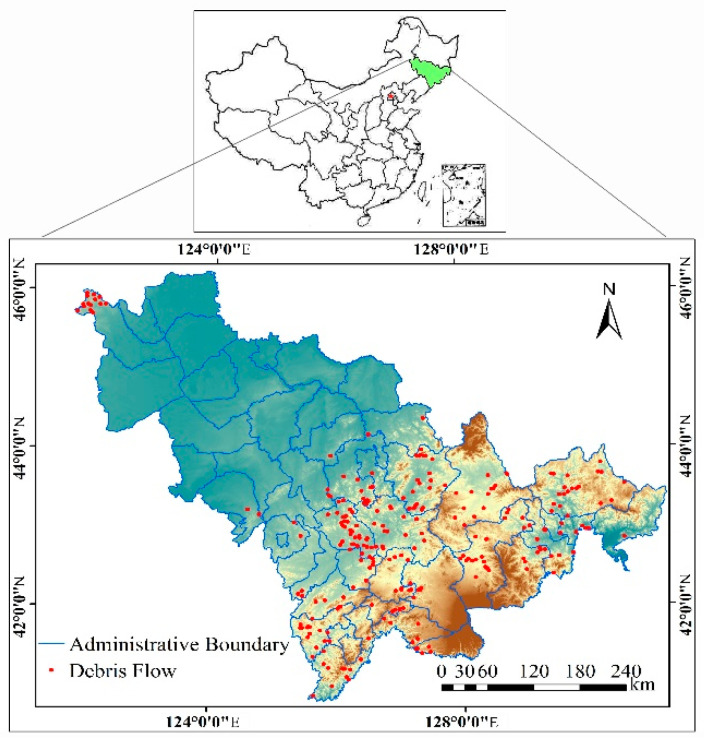
Location map of the study area.

**Figure 2 entropy-21-00695-f002:**
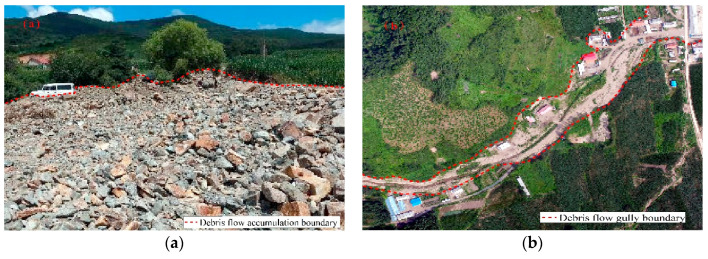
Debris flow field survey: (**a**) Debris flow accumulation; and, (**b**) The image of the debris flow ditch taken by drone.

**Figure 3 entropy-21-00695-f003:**
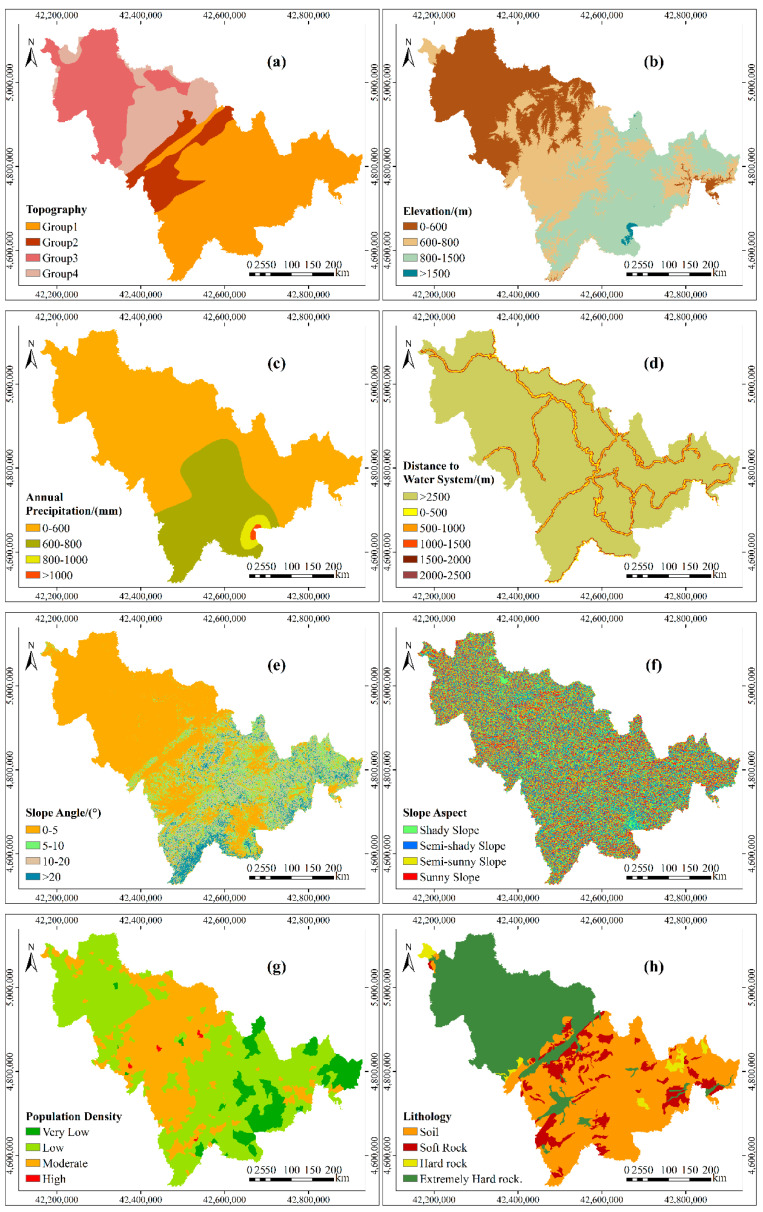
The hierarchical diagram of the controlling factors in the study area: (**a**) topography; (**b**) elevation; (**c**) annual precipitation; (**d**) distance to water system; (**e**) slope angle; (**f**) slopeaspect; (**g**) population density; (**h**) lithology; and, (**i**) vegetation coverage.

**Figure 4 entropy-21-00695-f004:**
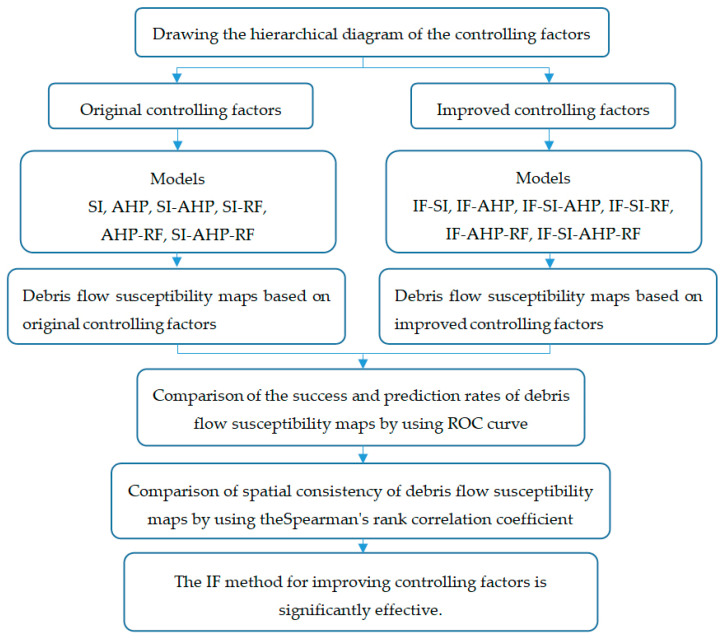
The flowchart of the research methods in this study.

**Figure 5 entropy-21-00695-f005:**
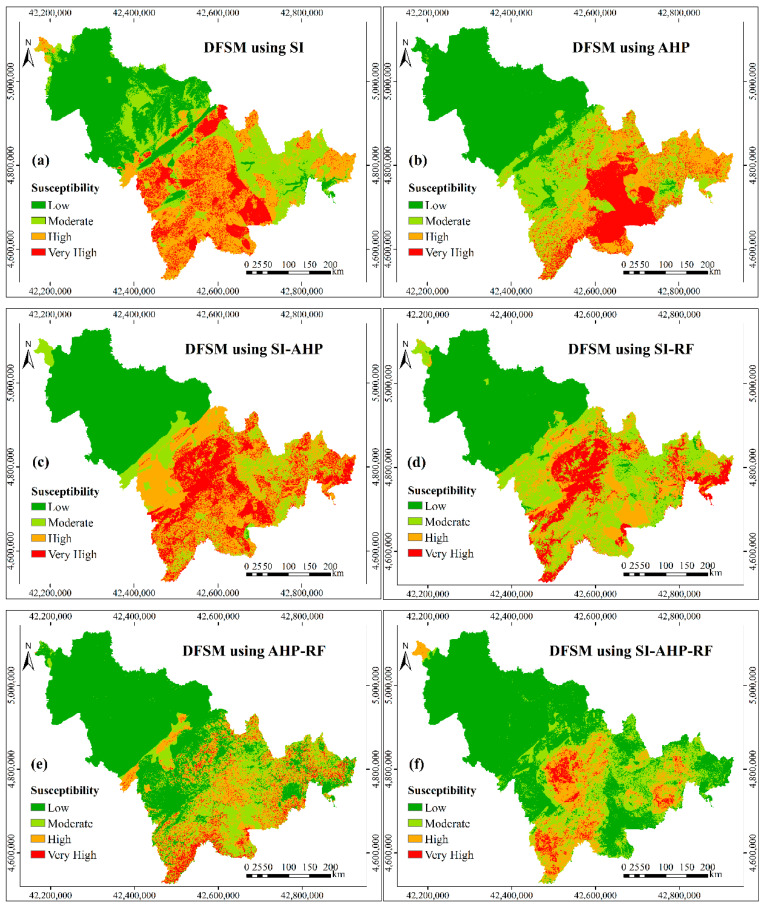
Debris flow susceptibility maps based on the controlling factors before information fusion (IF): (**a**) debris flow susceptibility mapping (DFSM)using statistical index (SI); (**b**) DFSM using analytic hierarchy process (AHP); (**c**) DFSM using SI-AHP; (**d**) DFSM using SI-random forest (SI-RF); (**e**) DFSM using AHP-RF; and, (**f**) DFSM using SI-AHP-RF.

**Figure 6 entropy-21-00695-f006:**
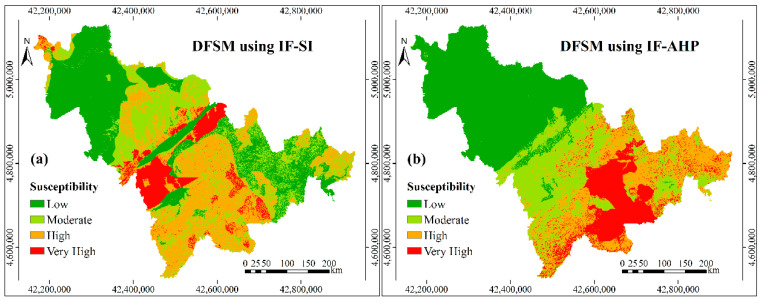
Debris flow susceptibility maps based on the controlling factors after IF: (**a**) DFSM using IF-SI; (**b**) DFSM using IF-AHP; (**c**) DFSM using IF-SI-AHP; (**d**) DFSM using IF-SI-RF; (**e**) DFSM using IF-AHP-RF; and, (**f**) DFSM using IF-SI-AHP-RF.

**Figure 7 entropy-21-00695-f007:**
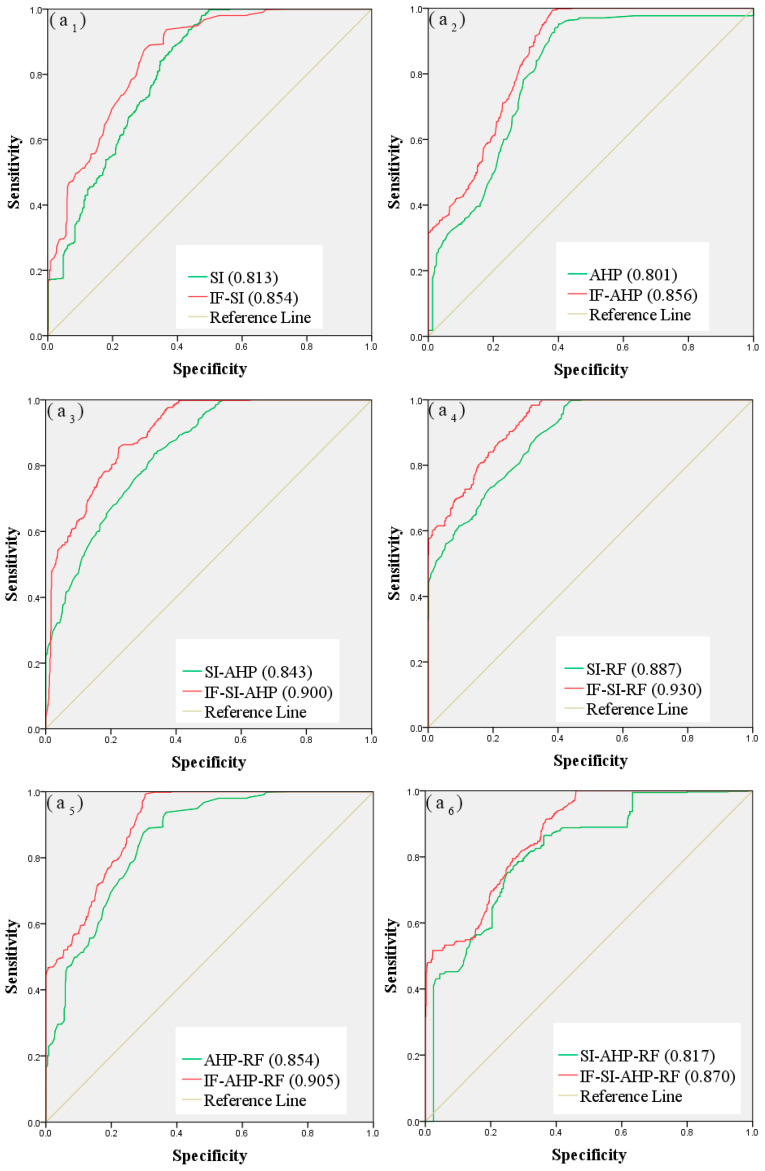
Comparison of success rate curves: (**a_1_**) SI and IF-SI; (**a_2_**) AHP and IF-AHP; (**a_3_**) SI-AHP and IF-SI-AHP; (**a_4_**) SI-RF and IF-SI-RF; (**a_5_**) AHP-RF and IF-AHP-RF; and, (**a_6_**) SI-AHP-RF and IF-SI-AHP-RF.

**Figure 8 entropy-21-00695-f008:**
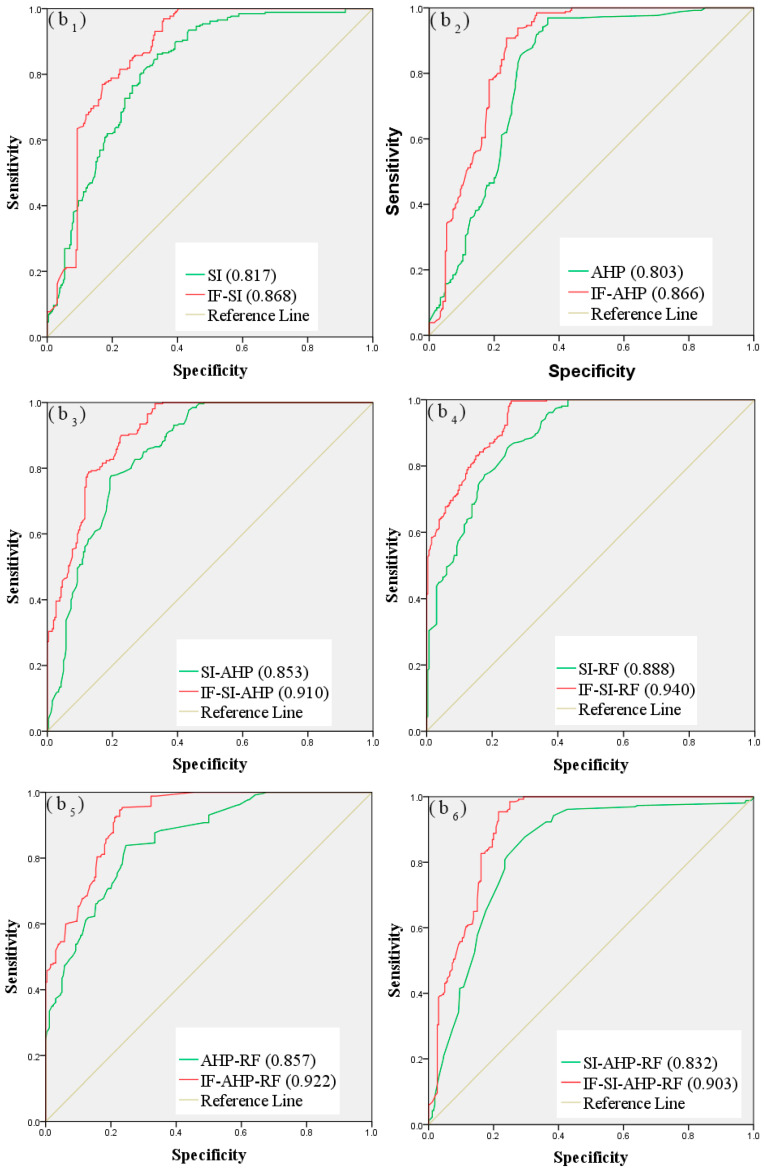
Comparison of prediction rate curves: (**b_1_**) SI and IF-SI; (**b_2_**) AHP and IF-AHP; (**b_3_**) SI-AHP and IF-SI-AHP; (**b_4_**) SI-RF and IF-SI-RF; (**b_5_**) AHP-RF and IF-AHP-RF; and, (**b_6_**) SI-AHP-RF and IF-SI-AHP-RF.

**Table 1 entropy-21-00695-t001:** Spatial database for the study area.

Data Layers	Data Type	Scale
Topography	Polygon	1:1,800,000
Elevation	Grid	30 m × 30 m
Annual precipitation	Polygon	1:5,000,000
Distance to water system	Polygon	1:5,000,000
Slope angle	Grid	30 m× 30 m
Slope aspect	Grid	30 m × 30 m
Population density	Polygon	1:1,800,000
Lithology	Polygon	1:1,800,000
Vegetation coverage	Polygon	1:1,800,000

**Table 2 entropy-21-00695-t002:** The Minkowski distance value.

Heading	X1^1^	X2^2^	X3^3^	X4^4^	X5^5^	X6^6^	X7^7^	X8^8^	X9^9^
X_1_	0	37.72	24.43	16.88	25.55	47.21	16.73	18.82	38.96
X_2_	37.72	0	35.81	19.29	29.43	40.32	40.09	37.96	64.24
X_3_	24.43	35.81	0	21.82	24.66	47.31	21.33	59.74	45.56
X_4_	16.88	19.29	21.82	0	21.68	52.38	4.58	63.73	35.17
X_5_	25.55	29.43	24.66	21.68	0	44.90	23.09	14.01	48.63
X_6_	47.21	40.32	47.31	52.38	44.90	0	52.58	35.45	70.32
X_7_	16.73	40.09	21.33	4.58	23.09	52.58	0	64.33	34.26
X_8_	18.82	37.96	59.74	63.73	14.01	35.45	64.33	0	85.53
X_9_	38.96	64.24	42.56	35.17	48.63	70.32	32.26	85.53	0
Support degree	266.3	324.86	277.66	255.53	271.95	390.47	254.99	459.57	422.67
Credibility degree	0.091	0.111	0.095	0.086	0.093	0.134	0.088	0.157	0.145

^1^ population density, ^2^ vegetation coverage, ^3^ lithology, ^4^ elevation, ^5^ slope angle, ^6^ slope aspect, ^7^ annual precipitation, ^8^ topography, ^9^ distance to water system.

**Table 3 entropy-21-00695-t003:** The standardized weights of the original controlling factors.

Controlling Factor	Class	SI	AHP	SI-RF	AHP-RF	SI-AHP	SI-AHP-RF
Topography	Group 1	−2.883	0.481	−2.029	1.335	−2.402	−1.548
	Group 2	0.464	0.481	1.318	1.335	0.945	1.799
	Group 3	−0.210	−0.440	0.644	0.414	−0.650	0.204
	Group 4	1.152	−0.652	2.006	0.202	0.499	1.353
Elevation(m)	0–600	0.764	−0.688	2.879	1.427	0.077	2.192
	600–800	−0.450	−0.688	1.665	1.427	−1.138	0.977
	800–1500	−2.224	−0.440	−0.109	1.675	−2.664	−0.549
	>1500	−0.390	−0.440	1.725	1.675	−0.830	1.285
Annual Precipitation(mm)	0–600	−1.217	−0.688	−1.457	−0.928	−1.904	−2.144
	600–800	1.101	−0.405	0.861	−0.645	0.697	0.457
	800–1000	0.168	0.162	−0.072	−0.078	0.330	0.090
	>1000	−1.459	1.437	−1.699	1.197	−0.022	−0.262
Distance to Water System (m)	>2500	0.101	−0.688	−1.092	−1.881	−0.587	−1.780
	0–500	1.022	1.225	−0.171	0.032	2.247	1.054
	500–1000	0.963	0.481	−0.230	−0.712	1.444	0.251
	1000–1500	0.911	−0.015	−0.282	−1.208	0.896	−0.297
	1500–2000	0.788	−0.369	−0.405	−1.562	0.419	−0.774
	2000–2500	0.683	−0.546	−0.510	−1.739	0.137	−1.056
Slope Angle (°)	0°–5°	−0.045	−0.298	0.220	−0.033	−0.343	−0.078
	5°–10°	0.608	0.268	0.873	0.533	0.877	1.142
	10°–18°	0.178	1.473	0.443	1.738	1.650	1.915
	>18°	−0.170	3.527	0.095	3.792	3.357	3.622
Slope Aspect	Shady Slope	−0.189	−0.440	−0.891	−1.142	−0.629	−1.331
	Semi-shady Slope	0.048	−0.652	−0.654	−1.354	−0.605	−1.307
	Semi-sunny Slope	0.252	−0.652	−0.450	−1.354	−0.401	−1.103
	Sunny Slope	0.446	−0.794	−0.256	−1.496	−0.348	−1.050
Population Density (people/km^2^)	Very Low	1.425	−0.263	1.536	−0.152	1.162	1.273
	Low	−0.344	−0.617	−0.233	−0.506	−0.961	−0.850
	Moderate	−1.055	−0.794	−0.944	−0.683	−1.849	−1.738
	High	−1.550	−0.865	−1.439	−0.754	−2.415	−2.304
Lithology	Soil	1.295	0.446	0.621	−0.228	1.741	1.067
	Soft rock	0.700	−0.298	0.026	−0.972	0.401	−0.273
	Hard rock	0.391	−0.688	−0.283	−1.362	−0.296	−0.970
	Extremely hard rock	−1.476	−0.688	−2.150	−1.362	−2.163	−2.837
Vegetation Coverage	Low	0.604	3.031	0.070	2.497	3.635	3.101
	Moderate	0.348	0.623	−0.186	0.089	0.971	0.437
	High	0.258	−0.263	−0.276	−0.797	−0.005	−0.539
	Very High	−1.008	−0.263	−1.542	−0.797	−1.271	−1.805

**Table 4 entropy-21-00695-t004:** The standardized weights of the improved controlling factors.

Controlling Factor	Class	IF-SI	IF-AHP	IF-SI-RF	IF-AHP-RF	IF-SI-AHP	IF-SI-AHP-RF
Topography	Group 1	−0.453	0.076	−0.319	0.210	−0.377	−0.243
	Group 2	0.073	0.076	0.207	0.210	0.148	0.282
	Group 3	−0.033	−0.069	0.101	0.065	−0.102	0.032
	Group 4	0.181	−0.102	0.315	0.032	0.078	0.212
Elevation(m)	0–600	0.066	−0.059	0.248	0.123	0.007	0.188
	600–800	−0.039	−0.059	0.143	0.123	−0.098	0.084
	800–1500	−0.191	−0.038	−0.009	0.144	−0.229	−0.047
	>1500	−0.034	−0.038	0.148	0.144	−0.071	0.111
Annual Precipitation(mm)	0–600	−0.107	−0.061	−0.128	−0.082	-0.168	−0.189
	600–800	0.097	−0.036	0.076	−0.057	0.061	0.040
	800–1000	0.015	0.014	−0.006	−0.007	0.029	0.008
	>1000	−0.128	0.126	−0.150	0.105	−0.002	−0.023
Distance to Water System(m)	>2500	0.015	−0.100	−0.158	−0.273	−0.085	−0.258
	0–500	0.148	0.178	−0.025	0.005	0.326	0.153
	500–1000	0.140	0.070	−0.033	−0.103	0.209	0.036
	1000–1500	0.132	−0.002	−0.041	−0.175	0.130	−0.043
	1500–2000	0.114	−0.054	−0.059	−0.227	0.061	−0.112
	2000–2500	0.099	−0.079	−0.074	−0.252	0.020	−0.153
Slope Angle(°)	0°–5°	−0.004	−0.028	0.021	−0.003	−0.032	−0.007
	5°–10°	0.057	0.025	0.081	0.050	0.082	0.106
	10°–18°	0.017	0.137	0.041	0.162	0.153	0.178
	>18°	−0.016	0.328	0.009	0.353	0.312	0.337
Slope Aspect	Shady Slope	−0.025	−0.059	−0.119	0.153	−0.084	−0.178
	Semi-shady Slope	0.006	−0.087	−0.088	−0.181	−0.081	−0.175
	Semi-sunny Slope	0.034	−0.087	−0.060	−0.181	−0.054	−0.148
	Sunny Slope	0.060	−0.106	−0.034	−0.200	−0.047	−0.141
Population Density(people/km^2^)	Very Low	0.130	−0.024	0.140	−0.014	0.106	0.116
	Low	−0.031	−0.056	−0.021	−0.046	−0.087	−0.077
	Moderate	−0.096	−0.072	−0.086	−0.062	−0.168	−0.158
	High	−0.141	−0.079	−0.131	−0.069	−0.220	−0.210
Lithology	Soil	0.123	0.042	0.059	−0.022	0.165	0.101
	Soft rock	0.066	−0.028	0.002	−0.092	0.038	−0.026
	Hard rock	0.037	−0.065	−0.027	−0.129	−0.028	−0.092
	Extremely hard rock	−0.140	−0.065	−0.204	−0.129	−0.206	−0.270
Vegetation Coverage	Low	0.067	0.336	0.008	0.277	0.403	0.344
	Moderate	0.039	0.069	−0.021	0.010	0.108	0.048
	High	0.029	−0.029	−0.031	−0.088	−0.001	−0.060
	Very High	−0.112	−0.029	−0.171	−0.088	−0.141	−0.200

**Table 5 entropy-21-00695-t005:** The Spearman’s rank correlation coefficients between the debris flow susceptibility maps.

Head	IF-SI	IF-AHP	IF-SI-RF	IF-AHP-RF	IF-SI-AHP	IF-SI-AHP-RF	SI	AHP	SI-RF	AHP-RF	SI-AHP	SI-AHP-RF
IF-SI	0	0.056	0.025	0.067	0.038	0.035	0.002	0.017	0.971	0.021	0.076	0.016
IF-AHP	0.056	0	0.670	0.013	0.864	0.016	0.071	0.001	0.021	0.032	0.780	0.056
IF-SI-RF	0.025	0.670	0	0.006	0.146	0.003	0.011	0.747	0.003	0.700	0.145	0.012
IF-AHP-RF	0.067	0.013	0.006	0	0.631	0.004	0.356	0.231	0.561	0.007	0.178	0.037
IF-SI-AHP	0.038	0.864	0.146	0.631	0	0.043	0.031	0.447	0.213	0.158	0.005	0.066
IF-SI-AHP-RF	0.035	0.016	0.003	0.004	0.043	0	0.063	0.039	0.059	0.231	0.321	0.004
SI	0.002	0.071	0.011	0.356	0.031	0.063	0	0.561	0.286	0.032	0.026	0.095
AHP	0.017	0.001	0.747	0.231	0.447	0.039	0.561	0	0.625	0.158	0.368	0.142
SI-RF	0.971	0.021	0.003	0.561	0.213	0.059	0.286	0.625	0	0.039	0.067	0.023
AHP-RF	0.021	0.032	0.700	0.007	0.158	0.231	0.032	0.158	0.039	0	0.159	0.067
SI-AHP	0.076	0.780	0.145	0.178	0.005	0.321	0.026	0.368	0.067	0.159	0	0.065
SI-AHP-RF	0.016	0.056	0.012	0.037	0.066	0.004	0.095	0.142	0.023	0.067	0.065	0

**Table 6 entropy-21-00695-t006:** Improvement of success rate and prediction rate.

Models	SI	AHP	SI-AHP	SI-RF	AHP-RF	SI-AHP-RF
Improved Success Rate	4.1%	5.5%	5.7%	4.3%	5.1%	5.3%
Improved Prediction Rate	5.1%	6.3%	5.7%	5.2%	6.5%	7.1%

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
