# Peer review of "A Method for Improving Controlling Factors Based on Information Fusion for Debris Flow Susceptibility Mapping: A Case Study in Jilin Province, China"

_entropy, 2019, doi:10.3390/e21070695_

Round 1

Reviewer 1 Report

The paper is quite difficult to read and the clarity of the English writing should be significantly improved before a further review. There are some details of the content either unclear or wrong and make it difficult to fully assess the technical merit of the approach. The deeper and clearer descriptions are necessary, but the narrative without related to this research department should be avoided as much as possible. This manuscript needs major revise in further because this article still with many problems need be clarified. Therefore, I have to reject this article in the current stage. Nevertheless, I still look forward to reviewing a resubmit manuscript which with English improvement and clearer writing in order to fully assess the technical merit of this work.

Author Response

Dear Reviewer:
Thanks for your letter and comments concerning our manuscript “A Method for Improving Controlling Factors Based on Information Fusion for Debris Flow Susceptibility Mapping: A Case Study in Jilin Province, China” (entropy-517507). Great appreciation goes to the editorial board and reviewer who kindly give excellent suggestions on writing and technical issues. We have revised this paper according to the reviewer’ suggestions and tried our best to perfect it. The structure of the text has also been modified to make it easier to read. A method to improve the factors controlling debris flow based on information fusion (IF) for debris flow susceptibility mapping is proposed and verified, and we take Jilin Province, China as the study area. Therefore, this text is divided into seven parts: Introduction, Study area, Data preparation, Methodology, Results, Discussion and Conclusions. As there are many changes in this manuscript, using the "Track Changes" function in Microsoft Word in the full text is confused. So I provided two revised manuscripts. One of them has used the "Track Changes" function to annotate the modifications in the article. In order to facilitate the review of the manuscript, another revised manuscript has highlighted the revisions in yellow in response to the suggestions given by the reviewer only. The main corrections in the paper and the responds to your comments are set out in the annex.

Reviewer 2 Report

Authors set nine layers of debris flow controlling factors, on the basis of experience and knowledge of the study area, and they were classified according to the previous study and existing classification. Since the choice of such controlling factors is of fundamental relevance in the applied method, I would suggest the Authors to give more details about this choice and to give some relevant references about previous study.

Line 142-157

The Authors stress the topography influence on geological hazards. It is not clear to the Reviewer how do they define the 4 different topographic groups, and how do they divide the 4 classes by elevation.

Line 158-161

The Authors put in evidence the rainfall effect on the debris flow occurrence, and they mention the concentrated rainstorms as the main cause of the most relevant debris flow events. Then, they refer to four classes by annual precipitation. The Reviewer ask the Authors to explain if they actually distinguish (and in case, how do they consider them in the study) between heavy rainfall and annual precipitation. It may reflect also on the discussion about rainfall influence (see line 380-384)

Figure 3, as it is reproduced in the manuscript, is not understandable. Higher quality is required.

Figure 4 reports “training sample Di” and “Classifier C1” which are not define into the text. It is opinion of the Reviewer that in this format figure 4 is almost useless. Therefore Reviewer suggests to cancel figure 4 or to change it.

Figure 5 and Figure 6 Low quality/resolution

Line 405-407

The sentence is not clear to the Reviewer. It would be appreciate if  Authors reformulate it.

Line 454-455

The sentence: “Due to the inevitable correlation between some controlling factors that affect debris flow,

the population density in the plain area will be relatively large, etc.” is not clear to the Reviewer. It would be appreciate if the Authors reformulate it

Minor remarks:

Line 47.

Missing reference for Wenbo Xu et al.

Line 90.

Misprinting in unit of measurement

Line 463

Misprinting. errata:DBSM - corrige: DFSM

Author Response

(The authors gave the same response as above.)

Reviewer 3 Report

The language of the article needs to be improved. 

Introduction section: "In addition, to better test the effect of the IF method, we selected methods that had been 78 recognized in the previous research from the three kinds of methods mentioned above..." Why can the integrated method make up for the shortcomings of a single method? Please explain. 

Why is the elevation divided into those categories? Please explain. 

"0~600 m, 600~800 m, 800~1500 m, and >1500 m".

Similar questions also apply to vegetation coverage classification and annual precipitation classification. 

The author applied several methods (e.g., AHP, Information fusion method, statistical index method) to the data to predict the percentage of debris flows. However, there is no justification for choosing the methods. The author didn't develop those methods further, so the innovation of the article is limited.

Author Response

(The authors gave the same response as above.)

Round 2

Reviewer 2 Report

Dear Authors,

I appreciated very much you considered all the comments I made on the original manuscript, and you provided a new version accounting for suggestions.

Every points have been straightforward considered and the new version of the manuscript correspond to the suggestions.

In my opinion, the manuscript it is worth to be published.

Kind regards